Liver proteome alterations in psychologically distressed rats and a nootropic drug

González-Fernández Raquel raquel.gonzalez@uacj.mx 1
Grigoruţă Mariana 1
Chávez-Martínez Sarahi 1
Ruiz-May Eliel 2
Elizalde-Contreras José Miguel 2
Valero-Galván José 1
Martínez-Martínez Alejandro 1
1 Departamento de Ciencias Químico Biológicas, Instituto de Ciencias Biomédicas, Universidad Autónoma de Ciudad Juárez , Ciudad Juárez , Chihuahua , Mexico
2 Red de Estudios Moleculares Avanzados, Instituto de Ecología A.C. (INECOL) , Xalapa , Veracruz , México
Chen Minjun
Electronic publication date: 2021 May 19
Publication date: 2021
Volume: 9
Electronic Location ID: e11483
Received 2020 Dec 15; Accepted 2021 Apr 27
Copyright: ©2021 González-Fernández et al.
Copyright year: 2021
Copyright holder: González-Fernández et al.
License: This is an open access article distributed under the terms of the Creative Commons Attribution License, which permits unrestricted use, distribution, reproduction and adaptation in any medium and for any purpose provided that it is properly attributed. For attribution, the original author(s), title, publication source (PeerJ) and either DOI or URL of the article must be cited.
License URL: https://creativecommons.org/licenses/by/4.0/

Keywords: Emotional stress, Distress, Liver proteome, Nootropics, Redox enzymes, NAFLD

Funding: CONACYT-México grant CB-2015 254483 SEP-PRODEP-Apoyo a Nuevos PTC UACJ-PTC-326 CONACYT-México CVU 548217 CONACYT-México U0004-20151 259915 This work was funded by the CONACYT-México grant CB-2015 (254483) to Alejandro Martínez-Martínez and SEP-PRODEP-Apoyo a Nuevos PTC (UACJ-PTC-326) to Raquel González-Fernández. Mariana Grigoruţă was supported by a doctoral fellowship from CONACYT-México (CVU 548217). The Orbitrap Fusion Tribrid Mass spectrometer was acquired by a grant to the CONACYT-México (U0004-20151, 259915). There was no additional external funding received for this study. The funders had no role in study design, data collection and analysis, decision to publish, or preparation of the manuscript.

==============================
Background

Chronic psychological distress is considered today a pandemic due to the modern lifestyle and has been associated with various neurodegenerative, autoimmune, or systemic inflammation-related diseases. Stress is closely related to liver disease exacerbation through the high activity of the endocrine and autonomic nervous systems, and the connection between the development of these pathologies and the physiological effects induced by oxidative stress is not yet completely understood. The use of nootropics, as the cognitive enhancer and antioxidant piracetam, is attractive to repair the oxidative damage. A proteomic approach provides the possibility to obtain an in-depth comprehension of the affected cellular processes and the possible consequences for the body. Therefore, we considered to describe the effect of distress and piracetam on the liver proteome.

Methods

We used a murine model of psychological stress by predatory odor as a distress paradigm. Female Sprague-Dawley rats were distributed into four experimental groups (n = 6 − 7/group) and were exposed or not to the stressor for five days and treated or not with piracetam (600 mg/kg) for six days. We evaluated the liver proteome by one-dimensional sodium dodecyl sulfate-polyacrylamide gel electrophoresis (1D-SDS-PAGE) followed by liquid chromatography-tandem mass spectrometry (GeLC-MS/MS). Besides, we analyzed the activity of liver antioxidant enzymes, the biochemical parameters in plasma and rat behavior.

Results

Our results showed that distress altered a wide range of proteins involved in amino acids metabolism, glucose, and fatty acid mobilization and degradation on the way to produce energy, protein folding, trafficking and degradation, redox metabolism, and its implications in the development of the non-alcoholic fatty liver disease (NAFLD). Piracetam reverted the changes in metabolism caused by distress exposure, and, under physiological conditions, it increased catabolism rate directed towards energy production. These results confirm the possible relationship between chronic psychological stress and the progression of NAFLD, as well as we newly evidenced the controversial beneficial effects of piracetam. Finally, we propose new distress biomarkers in the liver as the protein DJ-1 (PARK7), glutathione peroxidase 1 (GPX), peroxiredoxin-5 (PRDX5), glutaredoxin 5 (GLRX5), and thioredoxin reductase 1 (TXNDR1), and in plasma as biochemical parameters related to kidney function such as urea and blood urea nitrogen (BUN) levels.

Introduction

Psychological stress is considered a critical public health problem due to the modern lifestyle. When this tension is prolonged over time, it induces an emotional response referred to as distress characterized by a negative body response (Westley et al., 2021). This emotional condition causes physiological changes that involve both nervous and immune systems inducing systemic inflammation (Furman et al., 2019) that leads to mental illnesses, like anxiety, fear, and depression (Canteras, Pavesi & Carobrez, 2015). Nowadays, the connection between the mind and the body is well proved, which shows that physical wellness is closely bound to mental health. Thus, a dose–response relationship has been found between psychological distress and chronic diseases, such as cardiovascular diseases, hepatic disorders, and cancer (Russ et al., 2015; Yang et al., 2020).

The liver has an essential role in the body due to its function in the storage of glycogen, synthesis of plasma proteins, elimination of erythrocytes, detoxification, among all. Also, this organ has anabolic activity providing energy-rich compounds, such as glucose and lipids, essential for the adaptive response of body to distress (Duda et al., 2016). Increasing evidence indicates that psychological distress alters liver homeostasis. Chronic release of a high amount of glucocorticoids and catecholamines activates catabolic pathways in the liver and induces changes in the local immune response, which is closely related to the development or exacerbation of inflammatory liver diseases (Depke et al., 2009; Srivastava & Boyer, 2010). The impairment of the hepatic blood flow by the chronic liberation of corticotropin-releasing factor is also considered a mechanism through which emotional stimuli affect the liver (Chida, Sudo & Kubo, 2005). Thus, in previous studies, psychological distress caused changes in hepatic gene expression related to cell growth, proliferation and survival, local inflammatory response, and protein and lipid metabolism, and stimulated gluconeogenesis, hypercholesterolemia, and hepatic steatosis development (Adachi, Kawamura & Takemoto, 1993; Depke et al., 2008; Depke et al., 2009; Zhao et al., 2013). Furthermore, both physical and psychological distress induced oxidative stress, by altering antioxidant enzyme activity and glutathione level in hepatic tissue (Jafari et al., 2014).

The hypothalamic-pituitary-adrenal axis modulates the neurobehavioral and physiological responses to stress, triggering the release of reactive oxygen species (ROS) that cause oxidative damage to the DNA, proteins and lipids (Kinlein & Karatsoreos, 2019). The oxidative stress generated in response to psychological distress is related to various diseases such as neurodegenerative diseases (Wadhwa & Maurya, 2018), type 2 diabetes mellitus (Hackett & Steptoe, 2017), autoimmune diseases (Sharif et al., 2018), cancer (Kruk et al., 2019), among all. Although it is assumed that there is a relationship between the physiological effects of oxidative stress and inflammation caused by psychological distress and the development of such diseases, it is still unclear how this connection occurs. Moreover, endocrine regulation and changes in hormone levels across the lifespan can affect the mechanisms and consequences of the distress response. An increased susceptibility to autoimmune and other non-viral liver diseases (Krok & Koteish, 2010; Guy & Peters, 2013) and depression and anxiety disorders (Hanamsagar & Bilbo, 2016) was reported in women compared to men. However, female cyclical hormonal variations are often a reason to exclude them in research studies.

In previous studies, we showed that distress induces alterations in the redox status and in the bioenergetics in several brain regions like the amygdala, prefrontal cortex, and midbrain. Also, we associated these changes with anxiety, motor dysfunction (Mejia-Carmona et al., 2014; Grigoruţă et al., 2019), neuroinflammation and leukocyte recruitment in the area postrema (Vargas-Caraveo, Pérez-Ishiwara & Martínez-Martínez, 2015). Furthermore, this neuroinflammation is associated with lipid peroxidation and protein denaturation, measured by synchrotron radiation-based FTIR (SR-µFTIR), in circulating lymphocytes from rats exposed to predatory odor (Grigoruţă et al., 2018). Moreover, psychological distress reduced the mitochondrial content in the midbrain and the rate of glycolysis and oxidative phosphorylation in the prefrontal cortex and phenocopied some features of Parkinson’s disease (PD) neuropathology. Also, distress significantly changed the expression of the antioxidant enzymes in the brain: it induced the increase of SOD2 expression in the prefrontal cortex and the decrease of CAT level in the midbrain from male rats, and reduced SOD2 expression in the prefrontal cortex and SOD1 and SOD2 levels in the striatum in female rats (Grigoruţă et al., 2018).

Research on pharmacological compounds able to repair the oxidative damage caused by psychological stress is of great interest. The nootropics are known as “smart drugs” and stimulate memory and cognition. Piracetam was the first of these drugs to be described and, despite being a 2-oxo-1-pyrrolidine-acetamide, a derivate from gamma-aminobutyric acid (GABA), it showed opposite effects. It is commonly used as a neuronal enhancer because accumulated studies showed that it mainly modulates cholinergic and glutamatergic neurotransmission (Malykh & Sadaie, 2010). In clinical practice, piracetam is used as adjuvant therapy, as it protects against brain damage and improves cognitive functions in epilepsy and ischemia, and recently his antiparkinsonian activity was reported (Gupta et al., 2014). It is well known that piracetam protects neuronal cells against oxidative stress (Verma et al., 2018), increases mitochondrial membrane fluidity and ATP production (Müller et al., 1997), and improves the mitochondrial membrane potential (Keil et al., 2006; Kurz et al., 2010), as well as the glucose uptake and utilization (Abdel-Salam et al., 2013; Pandey & Garabadu, 2016), making this drug a metabolic enhancer and a neuroprotector. Thus, in animal and human studies, piracetam protected the brain against memory and cognitive function impairments induced by physical and chemical agents or by age (Kessler et al., 2000; Holinski et al., 2008; Gupta et al., 2009; Leuner et al., 2010; Abdel-Salam et al., 2011; Kosta et al., 2011; Muley et al., 2013; Wang, Li & Chen, 2016) and improved cognition and working memory in healthy volunteers (Alkuraishy et al., 2014). In a previous study, we also evidenced the anxiolytic effect and the protective role of this drug against lipid peroxidation in circulating mononuclear cells in rats subjected to psychological distress (Grigoruţă et al., 2018). However, there are few studies related to the effect of this drug on the liver. Piracetam had protective effects against the cytotoxicity of valproic acid in hepatocytes (Shrestha et al., 2014) but did not induce significant effects on oxidative stress markers from mice liver treated with cannabis extract and this drug (Abdel-Salam et al., 2013).

The analysis of liver response under anomalous conditions has traditionally been investigated through a reductionist vision. The development of technologies such as proteomics made holistic research possible for obtaining an overview of cellular responses under certain conditions. Proteomic studies carried out in the liver from murine models exposed to psychological stress are scarce, and the authors have mainly focused on brain structures (López-López et al., 2016). Previous proteomic analysis of liver from mouse under chronic unpredictable mild stress (CUMS) paradigm for five weeks showed alterations in proteins related to inflammation, immune regulation, lipid metabolism, and NF-κB signaling network (Wu et al., 2016).

Briefly, the liver is closely connected with the central nervous system, and in distress, both suffer significant changes. Both brain and liver are highly implicated in the maintenance the body homeostasis under stress conditions, and studies about the impact of oxidative stress on the brain-liver axis are very limited. Neuroprotective drugs, like piracetam, are used to diminish the effect of free radicals. Thereby, proteomic studies in the liver about the impact of this drug in distress are needed to evaluate also possible protective effects. The present study aimed to assess the consequence of distress and piracetam on the female rats’ health by the analysis of behavior, plasma biochemical parameters, liver proteome, and antioxidant enzyme activity.

Materials & Methods

Animals and experimental design

Twenty-eight healthy adult female Sprague-Dawley rats (Rattus novergicus) of 10–12 weeks of age, weighing 170 g to 270 g, were used. The animals were maintained in a controlled environment in light-dark cycle of 12/12 h, under room temperature conditions of 24 ± 2 °C, with food and water ad libitum. The animals were kept in pairs in standard plastic cages with no enriched environment. Five days before the experiment were housed individually to minimize the stress by separation. Rats were purchased in Rismart, S.A. de C.V., México.

The experimental design was carried out for six days. The induction of distress was performed in the first five days, and the rats were sacrificed on the sixth day. Each day, before distress exposure, rats belonging to the treatment group received piracetam and rats belonging to the control group received the same amount of water instead of the drug (Fig. S1A). The rats were randomly allocated by drawing lots into four experimental groups (n = 6–7 for each group): rats neither exposed to stressor nor with piracetam treatment (S−P−), rats exposed to stressor without piracetam treatment (S+P−), rats not exposed to stressor but treated with piracetam (S−P+), and rats exposed to stressor and treated with piracetam (S+P+) (Fig. S1C). No animal exclusions during the experiments were made in this study.

Administration of piracetam and distress induction

Piracetam (Nootropil® (1 g/5 mL), UCB Pharma Belgium) was administered as a single oral dose of 600 mg/kg in a final volume of 600 µL using a cannula, for six days, always at the same time (7:30 am) (Fig. S1A).

Every day, the induction of distress was carried out after the administration of piracetam from 8:00 am to 9:00 am. For it, a special box (60 × 27 × 35 cm) made of plastic was used, with the third part of black walls (the part of the box where the animals can hide, the black box) and two parts with transparent walls, separated by a black room divider with a small entrance through which the animal can pass (Dielenberg & McGregor, 2001) (Fig. S1B). Rats were left inside the boxes for 20 min to habituate to the new habitat. After this time, the animals were returned to their housing cage for 20 min to relax. Meanwhile, the box was conditioned by placing a 20 × 20 cm white cloth in the transparent part: a clean cloth was placed for the control group while, for the distressed group, the piece of cloth was previously placed in contact with a domestic cat for several days (as a cat rug). Next, rats were transferred to the experimental boxes for another 20 min. Each piece of cloth was stored in hermetic plastic bags and stored at −20 °C until its next use. The rats were videotaped for further behavior analysis using JWatcher™ v0.9 software, and the obtained data were expressed as the full time spent in each specific position each day (Grigoruţă et al., 2018).

All experiments were approved by the Institutional Committee of Ethics and Bioethics from the Universidad Autónoma de Ciudad Juárez, Ciudad Juárez, Chihuahua, México (No.CIBE-2017-1-34), following the Official Mexican Norm (NOM-062-ZOO-1999) and the ARRIVE guidelines 2.0 (Du Sert et al., 2020). No humane endpoints were needed in this study.

Sample collection

On the sixth day, animals were intraperitoneally anesthetized using sodium pentobarbital (45 mg/kg). After the loss of sensitivity and motor reflexes, whole blood samples were collected by cardiac puncture using vacutainer EDTA tubes for plasma biochemical parameter analysis. Further, rats were decapitated and, finally, the liver was extracted and weighted. One gram of liver was aliquoted for enzymatic activity assays and the rest of it was stored at −80 °C for proteomic analysis (Fig. S1C).

Blood plasma biochemical parameter analysis

The whole blood was centrifugated at 1,500 × g for 10 min. Then, the fresh plasma was collected and measured using a Cobas® c111 analyzer (Roche) for each biochemical parameter with the reagents provided by the manufacturer, following the manufacturer’s specifications (Fig. S1C).

Liver enzyme activity assays

Fresh liver aliquots (1 g) were homogenized (weight/volume 1:10) in cold 50 mM phosphate buffer (K2HPO4, KH2PO4, pH 7.4), with 1% phenylmethylsulfonyl fluoride (PMSF). After homogenization, the samples were sonicated three times for 30 s, with 1 min pause between each round, over the ice. Further, these samples were separated into aliquots for the different enzymatic activity methodology (n = 3–5). For catalase (CAT) activity analysis, the samples were centrifugated at 52 ×  g for 20 min at 4 °C. For the study of the enzymatic activity of superoxide dismutase (SOD), glutathione-S transferase (GST), glutathione peroxidase (GPX), and glutathione reductase (GR), the samples were centrifugated at 20,800 ×  g for 30 min at 4 °C, as previously described (Mejia-Carmona et al., 2014; Mejia-Carmona et al., 2015). In both cases, the supernatants were stored at −80 °C till use. Enzyme activity was measured by spectrophotometry using a microplate reader FLUOstar Omega (BMG) for 96-wells plates (Fig. S1C).

CAT activity was determined according to the method of Aebi (1984) with small modifications. As an oxidant medium was used 14 mM H2O2 (Sigma-Aldrich, H1009) in 0.1 M phosphate buffer (K2HPO4, KH2PO4, pH 7.4), and the sample was added in a final volume of 220 µL per well. Kinetic measurements were obtained at 240 nm (Mejia-Carmona et al., 2014). One unit of the enzyme represents the quantity of CAT used to neutralize one µM of peroxide per min.

SOD activity was determined by autooxidation of pyrogallol, a method developed by Marklund and Marklund (Marklund & Marklund, 1974). This method is based on the ability of pyrogallol for autooxidation, capturing the superoxide radicals at pH 8.2 and developing a yellow color that can be detected spectrophotometrically. SOD eliminates the radicals from de media and inhibits the pyrogallol oxidation. Total SOD and mitochondrial SOD (SOD2) activity were measured in 50 mM Tris–HCl, pH 8.2, containing 1 mM dietilentriaminopentaacetic acid (DTPA) as an oxidant medium. For the SOD2 activity, 110 mM sodium cyanide (NaCN) solution was also added to inhibit the specific activity of cytosolic SOD (SOD1). Finally, pyrogallol (Sigma 254002) solution was automatically injected by the microplate reader reaching a final concentration of 4 mM in a total volume of 300 µL per well, and kinetic measurements were obtained at 450 nm, as previously described (Mejia-Carmona et al., 2015). One unit of SOD was considered as the concentration of enzyme used to inhibit 50% of pyrogallol autooxidation. Specific enzyme activity of SOD1 was calculated as the difference between the enzyme activity of total SOD and SOD2.

The GST activity was determined according to the method of Habig, Pabst & Jakoby (1974) with some modifications. This method is based on the capacity of glutathione to combine with 1-chloro-2,4-dinitrobenzene (CDNB) to form the complex glutathione-CDNB. A 0.1 M Phosphate buffer (K2HPO4, KH2PO4, pH 6.5) containing 2.25 mM CDNB (Sigma, 237329) and 2.25 mM reduced glutathione (GSH, Sigma-Aldrich, G4251) was used as the substrate solution. Further, the sample was added in a final volume of 225 µL in each well. Kinetic measurements were obtained at 340 nm, as previously described (Mejia-Carmona et al., 2015). One unit of the enzyme activity was considered as the concentration of the GST necessary to obtain one µM of the complex glutathione-CDNB per min.

GPX activity was determined according to Weiss, Maker & Lehrer (1980) with some modifications. This method is based on the measurement of the disappearance of reduced nicotinamide adenine dinucleotide phosphate (NADPH). Fresh substrate solution was prepared in 0.1 M phosphate buffer (K2HPO4, KH2PO4, pH 7.4) containing 2 mM EDTA, 1.4 U glutathione reductase (Roche, 10105678001), 1 mM GSH, 0.2 mM NADPH, 0.5 mM H2O2 (Sigma-Aldrich, H1009), and 1 mM NaCN. The substrate solution and the sample were added to each well in a final volume of 300 µL. Kinetic measurements were obtained at 340 nm. One unit of enzyme activity represents the concentration of GPX that oxidizes one µM of NADPH per min.

The method used for the study of GR activity is based on the measurement of the reduction of NADPH, as previously published by Weiss, Maker & Lehrer (1980) with few modifications. The substrate solution was prepared from 20 mM oxidized glutathione (GSSG) (Abcam, ab141393) and 2 mM NADPH (Sigma-Aldrich, N5130) diluted in 0.1 M phosphate buffer (K2HPO4, KH2PO4, pH 7.4) with 2 mM EDTA. The substrate solution and the sample were added to each well in a final volume of 300 µL. Kinetic measurements were obtained at 340 nm. One unit of enzyme activity represents the concentration of GR that oxidizes one µM of NADPH per min.

Liver protein extraction

The remaining liver was lyophilized and grounded using a mortar and a pestle. For protein extraction, a TCA/acetone-based method was used. Briefly, 600 µL of cold precipitation solution (10% (w/v) TCA; 100% (v/v) acetone, 0.07% (w/v) DTT) was added on 40 mg of pulverized liver. Samples were sonicated 6 × 10 s at 30 W (6.9 kHz; Sonic Dismembrator, Model 100, Fischer Scientific) and let to precipitate at −20 °C overnight. Subsequently, the samples were centrifuged at 18,500  × g for 10 min at 4 °C and the supernatant was decanted. Then, pellets were twice washed using one mL of cold washing solution (80% (v/v) acetone, 0.07% (w/v) DTT) (González-Fernández et al., 2014). Finally, protein pellets were dried at room temperature to eliminate acetone residue. Proteins were solubilized using 100 µL of a solution containing 7 M urea, 2 M thiourea, 4% (w/v) CHAPS, 0.5% (w/v) Triton X-100, 20 mM DTT, 1 mM PMSF. Proteins were quantified using the Bradford assay (Bradford Reagent, B6916, Sigma-Aldrich®), according the manufacturer’s procedure. For the protein identification, 50 µg of total proteins extracted from each liver were pooled (n = 6 − 7) for each experimental group (Fig. S1C).

GeLC-MS/MS analysis

The protein identification was performed using one-dimensional sodium dodecyl sulfate-polyacrylamide gel electrophoresis (1D-SDS-PAGE) coupled by liquid chromatography-tandem mass spectrometry (GeLC-MS/MS) analysis (Fig. S1C). A total of 100 µg of pooled proteins from each experimental group was pre-separated by 1D-SDS-PAGE, gels were stained by the Coomassie method, and finally, each gel band was dissected manually in three equal parts. In-gel protein digestion and nano LC-MS/MS were performed according to the methodology previously described (Espinosa-Gómez et al., 2020) (for the complete description of the methodology carried out in this analysis, see Methodology S1). Briefly, firstly, each gel section was distained, and then dehydrated. Finally, proteins in gel sections were reduced with DTT and alkylated with iodoacetamide. For in-gel digestion, the gels sections were incubated in a solution containing 12.5 ng/µL mass spectrometry grade Trypsin Gold (Promega, Madison, WI, USA) in 5 mM NH4HCO3, at 37 °C overnight. Peptide samples were analyzed using an UltiMate 3000 RSnanoLC system (Thermo-Fisher Scientific, San Jose, CA) interfaced with Orbitrap Fusion™ Tribid™ (Thermo-Fisher Scientific, San Jose, CA) mass spectrometer. All MS data were obtained through Xcalibur 4.0.27.10 software (Thermo-Fisher Scientific) (Espinosa-Gómez et al., 2020). Each sample of pooled proteins for each experimental group was run on one time.

Proteomics data analysis and biological interpretation

Mass spectra were analyzed with Proteome Discoverer 2.1 software (PD, Thermo Fisher Scientific Inc.). The later searches were performed through Mascot server (version 2.4.1, Matrix Science, Boston, MA), SEQUEST HT (Eng, McCormack & Yates, 1994), and AMANDA (Dorfer et al., 2014). For protein identification, 25, 3.7, and 200 scores were considered, respectively, for each search engine. The search with each engine was conducted against the UniProt rat reference proteome (R. novergicus) database (https://www.uniprot.org/proteomes/UP000002494) (for the complete description of the protein search parameters in this analysis, see Methodology S1). The exponentially modified protein abundance index (emPAI) was calculated for the label-free relative quantitation of the proteins in each sample analyzed by nanoLC-MS/MS (Shinoda, Tomita & Ishihama, 2009). Only proteins identified with at least two unique peptides and a 1.5-fold difference in their relative abundance (0.66 ≤ fold change ≥ 1.50) were considered for further analysis.

Gene Ontology (GO) analysis and protein network were performed using the Search Tool for the Retrieval of Interacting Genes/Proteins (STRING) (https://string-db.org/cgi/input?sessionId=b8yhKk9BbQSn&input_page_show_search=on) (Szklarczyk et al., 2019). Enrichment pathway analysis was carried out using the Gene Annotation & Analysis Resource Metascape (http://metascape.org/gp/index.html#/main/step1) (Zhou et al., 2019). The enrichment cluster analysis was made using the following settings: p-value < 0.01, minimum count of 3, enrichment factor > 1.5, and FDR > 0.05.

Statistical analysis

The normality of the animal behavior, the antioxidant enzyme activity, and the biochemical plasma data were tested using the Kolmogorov-Smirnoff (KS) test. Furthermore, for the descriptive data, one-way ANOVA and Pearson’s correlation coefficient (rho) were determined, considering a statistically significant of p ≤ 0.05. The statistical analysis was performed using SPSS v.8.0 software (SPSS Inc. Chicago, IL, USA).

Results

Protein identification by GeLC-MS/MS approach

To evaluate the effect of the psychological distress and piracetam on female rat liver proteome, proteins were extracted using a TCA-acetone-based method. Protein samples from each liver were pooled (n = 6 − 7) and pools from each experimental condition were subjected to a pre-separation by 1D SDS-PAGE. Then, proteins were in-gel digested by trypsin, extracted from gel pieces and analyzed by a shotgun proteomic approach. Each experimental group was analyzed independently by label-free relative quantitation using the emPAI values. A total of 1,302 proteins were identified with at least one unique peptide and 1% FDR (Data S1). Of these, 894 proteins with at least two unique peptides were considered for further analysis. The intersection of proteins identified in all the experimental conditions was shown by a Venn diagram (Fig. 1A). A 1.5-fold change threshold was used as cut-off values for protein abundance changes (50% of value variation). According to this criterion, a total of 350 proteins exhibited qualitative differences between the four conditions: 112 proteins were found unique in one of the four experimental conditions, 103 were presented only in two of the four groups, and 135 in three groups (Fig. 1A). Regarding the quantitative differences, 313 proteins showed a change in their abundance comparing S+P− versus S−P−, 259 proteins in S+P− versus S+P+, and 315 proteins in S−P+ versus S−P− (Data S2).

Figure 1 Classification of proteins identified in the rat liver in each treatment by GeLC-MS/MS.

(A) Venn diagram comparing the number of proteins detected with at least two unique peptides for all experimental groups and the relationship between them. A total of 554 proteins were detected in the four experimental groups. However, 18, 35, 42, and 17 proteins were unique in S−P−, S−P+, S+P−, and S+P+, respectively. (B) GO analysis representing the subcellular localization of identified proteins performed using STRING webtool. Most of the detected proteins in the four experimental groups were located within membrane, cytosol, mitochondria, and nucleus. Protein number is referred to as the count in a gene set, in this case, in the GO-term. S−P−, rats neither exposed to stress nor with piracetam treatment; S−P+, rats not exposed to stress but treated with piracetam; S+P−, rats exposed to stress without piracetam treatment; S+P+, rats exposed to stress and treated with piracetam.

S−Pntology through STRING search tool annotation was used to determine the subcellular localization of all proteins identified. The most highly represented categories in this study were membrane-associated, mitochondrial, cytosolic, and nuclear proteins, counting more than 100 proteins in each category (Fig. 1B).

Figure 2 Pathway enrichment analysis of altered proteins in the liver of rats exposed to distress and piracetam.

The vertical axis indicates the top significant terms in KEGG pathway enrichment analysis of the most and less abundant proteins from (A) S+P− compared to S−P−, (B) S+P+ compared to S+P−, and (C) S−P+ compared to S−P−. Proteins related to amino acids metabolism, glucose oxidation, fatty acid mobilization and degradation, protein folding, trafficking and degradation, metabolism of xenobiotics by cytochrome P450, GSH metabolism, antioxidant defense mechanisms, and non-alcoholic fatty acid liver, among others, were highly affected in distress (S+P−) when compared to the control group (S−P−). Piracetam reverted the changes in metabolism caused by distress. Piracetam administration to non-distressed rats (S−P+) induced changes in proteins involved in amino acid, glucose, and fatty acid metabolism, as well as in the metabolism of GSH and xenobiotics by cytochrome P450, among others, compared to the control group (S−P−). Proteins that showed a 1.5-fold change in their relative abundance were considered for the analysis. The analysis was made using Metascape resource, providing the significantly enriched KEGG pathway terms across input protein lists (−log10 p-value; horizontal axis).

Biological interpretation of differentially accumulated proteins

Proteins function was analyzed using Kyoto Encyclopedia of Genes and Genomes (KEGG) annotation through Metascape online tool (Fig. 2). Liver from rats exposed to distress (S+P−) showed significant enrichment in 16 pathways in which proteins were less abundant, and in 31 pathways in which proteins were more abundant compared to the liver from rats not exposed to distress (S−P−) (Fig. 2A). In general, proteins related to amino acids metabolism, glucose and fatty acid (FA) mobilization and degradation to produce energy (glycolysis/gluconeogenesis, TCA cycle, pyruvate metabolism, FA degradation, PPAR signaling pathway, dicarboxylate metabolism, butanoate metabolism), protein folding, trafficking and degradation, metabolism of xenobiotics by cytochrome P450, GSH metabolism, antioxidant defense mechanisms, and non-alcoholic fatty acid liver or Parkinson’s diseases, among others, were highly affected in distress (S+P−) when compared to S−P− (Fig. 2A, Data S3).

When piracetam was supplied to the distressed group (S+P+), 24 and 19 pathways showed a significant enrichment in which proteins were less and more abundant, respectively, compared to S+P− (Fig. 2B). Piracetam reverted the changes in metabolism caused by distress exposure (Figs. 2A and 2B, Data S3). When piracetam was administrated to non-distressed rats (S−P+), 16 and 34 pathways showed a significant enrichment in which proteins were less and more abundant, respectively, compared to the control group (S−P−) (Fig. 2C). The functional enrichment analysis showed that proteins involved in amino acid metabolism, both biosynthesis and especially degradation pathways, and in the oxidation of glucose and fatty acids were more abundant. In the same way, proteins involved in the metabolism of xenobiotics by cytochrome P450 and GSH showed an increase in their abundance (Figs. 2A and 2C, Data S3).

This study highlights that psychological distress and piracetam induce changes in protein abundances that are involved in redox metabolism, non-alcoholic fatty liver disease (NAFLD), ER stress, and metabolism of xenobiotics by Cyt P450.

Altered proteins involved in redox metabolism

In this study, a total of 26 proteins involved in redox homeostasis were detected in the hepatic proteome, of which 19 of them presented differences in 50% in protein abundance changes between S+P− versus S−P−, S+P+ versus S+P−, and S−P+ versus S−P− (0.66 ≤ fold change ≥ 1.50) (Table S1).

To validate the changes in redox enzymes found in the proteomic analysis, the activity of the selected antioxidant enzymes was also measured by biochemical methods (Table 1, Data S4). Although no change in GPX protein abundance was shown in distressed rats (Table S1), a statistically significant increase about two-fold in the GPX activity was observed in this condition (p = 0.026) (Table 1), and piracetam decreased this effect by 10%, but without resetting the control group values (Table 1 and S1); however, the drug increased the protein abundance in 75% in the distressed rats (S+P+) (Table S1). When piracetam was administered to non-distressed animals (S−P+) also induces a statistically significant rise of more than 70% in GPX activity.

Table 1 Activity measure of redox enzymes expressed as IU per mg of fresh liver weight (FLW) in each experimental group.

Enzyme activity
(IU/mg LFW)	S−P−	S−P+	S+P −	S+P+	p-value	
CAT (x106)	221.6 ± 19.7c	167.1 ± 9.6a	228.5 ± 27.0c	182.4 ± 14.8ab	0.044	
GR	3.6 ± 0.4a	3.2 ± 0.4a	2.4 ± 0.2a	3.1 ± 0.3a	0.380	
GPX	5,839.0 ± 1,020.8a	5,997.9 ± 659.0ab	9,516.6 ± 1,801.7c	8,194.9 ± 1,383.5bc	0.025	
Total GST	223.6 ± 7.9a	220.6 ± 7.7a	227.3 ± 6.7a	230.9 ± 8.3a	0.454	
Total SOD	3,304.8 ± 179.0a	3,320.6 ± 67.8a	3,227.8 ± 71.3a	3,378.6 ± 199.5a	0.566	
SOD1	2,236.3 ± 182.3a	2,356.1 ± 67.3a	2,169.1 ± 137.3a	2,238.2 ± 258.8a	0.776	
SOD2	1,068.5 ± 73.3a	970.3 ± 49.9a	1,058.6 ± 82.5a	1,140.4 ± 61.4a	0.112	
Notes.

Data are represented as mean ± S.E.M. (n = 3–5/group). Different letters mean significative differences (ANOVA, p ≤ 0.05).

CAT catalase

GR glutathione reductase

GPX glutathione peroxidase

GST glutathione S-transferase

SOD superoxide dismutase

SOD1 Cu,Zn-SOD, cytosolic

SOD2 Mn-SOD, mitochondrial

S−P− rats neither exposed to stress nor with piracetam treatment

S−P+ rats not exposed to stress but treated with piracetam

S+P− rats exposed to stress without piracetam treatment

S+P+ rats exposed to stress and treated with piracetam

An increase of 50% in CAT protein abundance was observed in rat liver exposed to stress stimuli (Table S1); however, no significant changes in CAT activity were shown in this group (Table 1). Piracetam induced a statistically significant decrease of 20% in the CAT activity in distressed group (S+P+), and this reduction was about 25% in non-distressed animals (S−P+) (p = 0.044) (Table 1). The drug enhanced the CAT abundance by 18% in the distressed group (S+P+) compared to S+P− group and induced its increase for about 40% in non-distressed animals (S−P+).

Distress and piracetam induced no changes in total GST activity (Table 1); however, ten GST isoforms were found in the proteomic analysis with different trend in protein abundance changes (Table S1). Glutathione S-transferases, specifically glutathione S-transferase A4-4 (GSTA4), keep in balance the cellular 4-hydroxynonenal (4-HNE) concentration, an end-product of n-6 PUFAs peroxidation related to increased apoptosis and necrosis, through its conjugation to GSH (Singhal et al., 2015). In this study, the GSTA4 was found 50% less abundant in the distressed group (S+P−); however, piracetam produced an increase of two-fold in the abundance of this protein in distressed rats (S+P+) (Table S1).

Also, GR and SOD (total SOD, SOD1, SOD2) showed no activity changes induced by distress in the liver (Table 1). In LC-MS/MS analysis, GR and SOD2 abundance were not detected but SOD1 abundance was found 50% decreased in S+P− compared to S−P−. Piracetam decreased its abundance in animals under distress exposure (with 30%, S+P+ versus S+P−) and induced a decrease even more in no-distressed rats (with 70%, S−P+ versus S−P−) (Table S1).

Altered proteins involved in non-alcoholic fatty liver disease (NAFLD)

One of the pathways affected by distress and piracetam, in the present study, was related to NAFLD (rno04932) (Fig. 2). A total of 11 and 9 proteins were increased and decreased, respectively, under distress exposure (Table S2). Proteins from NADH dehydrogenase (Complex I), Cytochrome b-c1 complex (Complex III), response to oxidative stress, and apoptosis regulation were modified by the distress effect and this tendency was inverted by piracetam (Table S2).

Altered proteins involved in ER stress

The endoplasmic reticulum (ER) is an essential organelle involving in protein folding and trafficking, calcium homeostasis and is widely developed in hepatocytes. In this study, the protein processing in ER (rno04141) was affected by distress exposure (S+P−), causing a change in the abundance of proteins involved in the ER quality control by the retention of incorrectly folded proteins, in the ubiquitin-proteasome system, autophagy, and ER-associated protein degradation pathway, and in the protein processing, transport and secretion (Fig. 2A, Table S3). Piracetam provoked a shift in the abundance of proteins that were found less abundant in the liver of distressed animals (S+P+) (Fig. 2B, Table S3) but had no effect on those that were more abundant; moreover, the drug had no effect on non-distressed animals (S−P+) (Fig. 2C, Table S3). Besides, the ER stress could be assisted by the activation of the peroxisome proliferator-activated receptors (PPARs) signaling pathway (rno03320), which has been shown to mediate hepatic protection (Zhang et al., 2016). In this study, the PPAR signaling pathway was increased in distressed animals (S+P−) and by piracetam (in S+P+ and S−P+ groups versus S−P− group) (Fig. 2, Table S3).

Altered proteins involved in the metabolism of xenobiotics by Cyt P450

The biotransformation process involving Phases I and II enzymes of xenobiotics and drug metabolism was also affected by exposition to the stressor (rno00980) (Fig. 2A, Table S3). One of the proteins that turned out to be the most abundant in the liver of stressed rats was the cytochrome P450 2E1 (CYP2E1), which has been associated with steatohepatitis development. In this study, this protein was more abundant in animals exposed to distress (S+P−). Piracetam induced its decline in distressed rats (S+P+) but increased it in non-distressed ones (S−P+) compared to the control group (S−P−) (Figs. 2B and 2C, Table S3).

Interaction network of differently accumulated proteins

A Markov Clustering (MCL) algorithm was applied to identified proteins with abundance changes involved in PD (rno05012), NAFLD (rno04932), selected antioxidant enzymes (included in Table 1), PPAR signaling pathway (rno03320), processing in ER (rno04141), proteasome (rno03050), cytochrome P450 metabolism (rno00980), and selected proteins involved in apoptosis (caspase 6 (CASP6) and apoptosis-inducing factor 1, mitochondrial (AIFM1)) to show the associations among them, using de STRING protein-protein interaction database. These proteins were grouped into sixteen clusters (Fig. 3, Data S5). The protein interaction network showed a close relationship among these pathways. These clusters can be grouped into five sets and are distributed around a central group with proteins involved in redox status homeostasis as DJ-1, SOD1, CAT, GLRX5, PRDX2, PRDX5, PRDX6, and TXNRD1 (proteins in cluster 6, colored like green; see cluster information in Data S5).

Figure 3 Interaction network of selected proteins with changes in their abundance in the liver of rats exposed to distress and piracetam.

Proteins involved in redox metabolism, GSH metabolism, PD, NAFLD, PPAR signaling pathway, protein processing in ER, metabolism of xenobiotics by Cyt P450, and apoptosis were grouped into sixteen clusters using STRING database. The protein interaction network showed a close relationship among these pathways. Clusters were grouped into five sets distributed around a central group with proteins involved in redox status homeostasis. Edges represent protein-protein associations made using STRING database with a medium confidence level (0.4). Protein clustering was carried out by the Markov Cluster Algorithm (MCL) with a specific inflation parameter of 3. The different colors in circles indicate diverse clusters. Solid lines represent protein interactions that belong to the same cluster. Dashed lines represent protein interactions belonging to a distinct cluster. The line thickness shows the strength of evidence, with thicker connections presenting higher confidence in the protein-protein interaction (the detailed description of the clusters is shown in Table S4). The colored ellipses show the regrouping of the closest clusters.

Effect of chronic psychological distress and piracetam on animal behavior

Four of the rat behaviors were more common during the 20 min exposure to distress each day and were expressed as the time spent in each position (Dielenberg & McGregor, 2001; Grigoruţă et al., 2018) (Fig. 4, Data S4): two defensive behaviors that included hide, referring to the time the rat spent in the hiding place (Fig. 4A), and head-out, like the time the rat is spending in alarm position with the head out or with half of the body out from the hiding place, observing the horizon (Fig. 4B); and two non-defensive behaviors that included exploration, like the time the rat spent exploring the transparent part of the box (Fig. 4C), and approach, like the time the rat spent smelling the piece of cloth with or without cat odor stimuli (Fig. 4D). Rats exposed to cat odor stimuli (S+P−) were concealed in the hiding place during almost all the time of the distress exposure in all five days (97% of the total time) (Fig. 4A), comparing to non-distressed rats (S−P−) (p ≤ 0.00001), that spent more time exploring the open place (4%, p ≤ 0.0001), in head out position (12%, p ≤ 0.0000001), and approaching the cloth (5%, p = 0.032), from the second day (Figs. 4B, 4C, 4D). These observations showed the efficiency of the distress induction model by cat odor exposure.

Figure 4 Effect of chronic psychological distress and piracetam in rat behavior.

(A) Time spent hiding (Hide); (B) time spent in the entrance of the hiding place (Head-out); (C) time spent exploring the opened part of the box (Exploration); (D) time spent examining the cloth piece permeated (or not) with cat odor (Approach). Rats were exposed to cat odor as a stressor for five consecutive days. Distressed rats (S+P+) were in the hiding place during almost all the time of the distress exposure in all five days comparing to the non-distress animals (S−P−), that spent more time exploring the open place, in head out position, and approaching the cloth, from the second day. Piracetam did not change the behavior in distressed rats (S+P+), but in non-distressed rats (S−P+), the drug provoked an increase in the time spent in the concealment, and a decrease in the time used to explore, to approach the cloth and to be in the alert. S−P−, rats neither exposed to stress nor with piracetam treatment; S−P+, rats not exposed to stress but treated with piracetam; S+P−, rats exposed to stress without piracetam treatment; S+P+, rats exposed to stress and treated with piracetam. Data are shown as mean ±  S.E.M. (n = 5/group).

Piracetam did not change the behavior in distressed rats (S+P+) (Fig. 4). However, when the drug was administrated to non-distressed rats (S−P+), significant changes were observed compared to the control group (S−P−), increasing of 78% to 93% the time spent in the concealment, while the time used to explore, to approach the cloth and to be in the alert was decreased, as happened in S+P− group (Fig. 4).

Effect of chronic psychological distress and piracetam on plasma biomarkers

Several biochemical parameters including glucose, triglycerides, and cholesterol and other biomarkers that indicate hepatic injury and function (direct and indirect bilirubin, alanine aminotransferase, aspartate aminotransferase, total protein, and albumin levels) and kidney function (creatinine, urea, and blood urea nitrogen (BUN) levels) were measured in plasma in all experimental groups (Table 2, Data S4). The exposure to distress induced a statistically significant decrease in glucose levels in the distressed group (S+P−) (p = 0.012), and the administration of piracetam avoided such an effect. The same happened in the triglycerides level, albeit not significant changes. The specific biomarkers for hepatic injury and function were not affected by distress and/or piracetam, although the total bilirubin presented a tendency to decrease in distressed animals (S+P−) and with piracetam (S+P+) (p = 0.065). Moreover, a statistically significant increase in the BUN level was found in distressed animals (S+P−), and piracetam counteracted this effect reducing such levels, even more in rats exposed to distress (S+P−) (p = 0.025). Also, piracetam significantly decreased plasma urea levels in distressed rats (S+P+) compared to untreated rats (S+P−) (p = 0.025).

Correlational analysis among treatments, behavior, biochemical parameters, and antioxidant enzyme activity

To study the association between treatments and behavior, a normality test of the variables was first performed using Kolmogorov-Smirnoff (KS). Normal data distribution was shown and therefore the statistical test Pearson’s correlation coefficient (rho) was used, considering a statistically significant correlation p ≤ 0.05. A high correlation was found between treatments and behaviors (Table S4). The time spent hiding was positively correlated with the treatments (r = 0.82, p ≤ 0.00001), while the time spent in head-out position, exploring and approaching the piece of cloth were negatively correlated with the treatments (r =  − 0.73, r =  − 0.83, and r =  − 0.72, respectively; p ≤ 0.0001) (Fig. S2). These correlations showed that piracetam had no effect on distressed rats but induced defensive behavior in non-distressed rats (Grigoruţă et al., 2018).

Table 2 Delta body and liver weight and biochemical parameters in plasma obtained from each experimental group.

	S−P−	S−P+	S+P−	S+P+	p-value	
Delta weight (g)*	11.74 ± 2.13a	8.68 ± 3.94a	10.54 ± 3.97a	15.20 ± 7.12a	0.382	
Liver (g/100 g body weight)†	4.25 ± 0.15a	4.21 ± 0.19a	4.40 ± 0.23a	4.04 ± 0.18a	0.501	
Glucose (mmol/L)	14.49 ± 0.92b	11.52 ± 1.57a	12.66 ± 1.81a	14.96 ± 1.35b	0.012	
Triglycerides (mmol/L)	1.19 ± 0.11a	1.41 ± 0.40a	0.86 ± 0.04a	1.39 ± 0.27a	0.522	
Cholesterol (mmol/L)	1.68 ± 0.10a	1.69 ± 0.09a	1.64 ± 0.07a	1.68 ± 0.14a	0.983	
Liver injury						
Direct bilirubin (mmol/L)	0.46 ± 0.11a	0.37 ± 0.09a	0.32 ± 0.08a	0.38 ± 0.09a	0.514	
Indirect bilirubin (mmol/L)	1.27 ± 0.24a	0.89 ± 0.13a	0.99 ± 0.36a	0.67 ± 0.30a	0.120	
Total bilirubin (mmol/L)	1.73 ± 0.25a	1.40 ± 0.12a	1.33 ± 0.23a	1.22 ± 0.38a	0.067	
AST (IU/L)	50.40 ± 2.24a	51.60 ± 3.74a	55.72 ± 8.29a	54.50 ± 3.71a	0.683	
ALT (IU/L)	34.17 ± 2.73a	33.28 ± 1.51a	34.62 ± 3.40a	35.35 ± 2.92a	0.965	
Albumin (g/L)	36.47 ± 0.65a	36.39 ± 0.54a	35.74 ± 1.15a	35.94 ± 1.04a	0.714	
Total protein (g/L)	65.17 ± 3.19a	66.91 ± 3.42a	64.17 ± 3.43a	66.27 ± 4.12a	0.848	
Kidney function						
Urea (mmol/L)	5.46 ± 0.33ab	5.45 ± 0.42ab	5.65 ± 0.35b	4.68 ± 0.28a	0.025	
Creatinine (mmol/L)	24.71 ± 1.38a	20.02 ± 2.07a	22.88 ± 2.40a	22.37 ± 2.06a	0.178	
BUN (mmol/L)	2.59 ± 0.14b	2.52 ± 0.20ab	2.64 ± 0.17c	2.19 ± 0.13a	0.025	
Notes.

Data are represented as mean ± S.E.M. (n = 3–7/group). Different letters mean significative differences (ANOVA, p ≥ 0.05).

* Delta weight was calculated as the rats body weight on the fifth day minus its body weight on the first day of the experiment and is shown in absolute value.

† Liver value was calculated as the rats liver weight (g) per 100 g of rats body weight on the fifth day of the experiment.

AST aspartate aminotransferase

ALT alanine aminotransferase

BUN blood urea nitrogen

S−P− rats neither exposed to stress nor with piracetam treatment

S−P+ rats not exposed to stress but treated with piracetam

S+P+ rats exposed to stress without piracetam treatment

S+P+ rats exposed to stress and treated with piracetam

Further, correlational studies among biochemical parameters, antioxidant enzyme activity, treatments, and rat behavior were performed. The indirect and total bilirubin were negatively correlated with the treatments (r =  − 0.63, p = 0.02 and r =  − 0.58, p = 0.01, respectively) (Table S4). Moreover, glucose level was negatively correlated with the time spent hiding (r =  − 0.49, p = 0.04) (Fig. S2A, Table S4), while it was positively correlated with the time spent in head out position (r = 0.57, p = 0.04) (Fig. S2B, Table S4); direct bilirubin level was positively correlated to the time spent approaching and exploring (r = 0.67, p = 0.01 and r = 0.59, p = 0.03, respectively) (Figs. S2C, S2D, Table S4); and total bilirubin level was negatively correlated with the time spent hiding (r =  − 0.45, p = 0.05) (Fig. S2E, Table S4), and positively correlated with the time spent exploring (r = 0.45, p = 0.05) (Fig. S2F, Table S4). These correlations between behavior and biochemical parameters show that bilirubin levels are related to stress behavior and can be useful as a biomarker of psychological distress.

The correlational analysis among antioxidant enzyme activity with rat behavior and treatments showed that only GPX activity had a positive association with the treatments (r = 0.61, p = 0.01) (Table S4) and with the time spent hiding (r = 0.55, p = 0.01) (Fig. S2G), and it had negative association with the exploration time (r =  − 0.61, p = 0.01) (Fig. S2H). These correlations support the relationship of this enzyme with the stress behavior of hiding and it can be considered in future studies as a distress biomarker.

Discussion

The connection between the physiological effects induced by oxidative stress and inflammation caused by psychological distress and the associated pathologies is still unclear how occurs, as well as the questionable beneficial effects of piracetam. The proteomic analysis provides a global view of the affected cellular processes and the possible consequences on the body. Thus, a GeLC-MS/MS analysis followed by a label-free relative quantitation based emPAI values of identified proteins were carried out. Overall, our results showed that distress affected a wide range of proteins involved in amino acid metabolism, glucose, and fatty acid (FA) mobilization and degradation to produce energy, protein folding, trafficking and degradation, redox metabolism, and NAFLD development. Piracetam reverted the changes in metabolism caused by distress exposure, and, under physiological conditions, the drug induced an increased catabolism rate directed towards energy production.

Physiologically, the body’s primary response to stress is the fight or flight response that, at the metabolic level, involves an increase of the energy production rate (increase ATP). After exposure to a stressor, the hypothalamic-pituitary-adrenal axis is activated, causing the production of the so-called stress hormones, such as epinephrine and cortisol (corticosterone in murine), which in turn cause an increase of glycogenolysis, gluconeogenesis, lipid mobilization, and inhibition of protein synthesis. Also, ATP synthesis leads to ROS generation, which can promote oxidative stress, causing cellular damage to DNA, lipids, and proteins (Schiavone et al., 2013; Dumbell, Matveeva & Oster, 2016). Previous studies showed that emotional stress induces oxidative damage and altered metabolism in the liver in different murine psychological stress models (Depke et al., 2008; Depke et al., 2009; Jafari et al., 2014).

In the present study, we focused on liver proteins involved in redox metabolism, NAFLD, ER stress, and metabolism of xenobiotics by Cyt P450, which were changed by psychological distress and piracetam.

Redox metabolism

The maintenance of optimal ROS levels is essential in the body due to the dual function of these species. ROS are signaling molecules that activate transcription factors that regulate the gene expression related to growth and cell differentiation, but, on the other hand, ROS excess has been associated with the development of numerous diseases such as type 2 diabetes, autoimmune diseases, cancer or neurodegenerative diseases, among all (Hackett & Steptoe, 2017; Wadhwa & Maurya, 2018; Sharif et al., 2018; Kruk et al., 2019). The relationship between psychological distress and the redox status alteration of the body has been previously shown in several studies using different murine models in the liver, brain, pancreas, kidney, lungs, and heart (Şahin & Gümüşlü, 2007; Jafari et al., 2014; Mejia-Carmona et al., 2014; Mejia-Carmona et al., 2015; López-López et al., 2016; De Sousa Rodrigues et al., 2017).

In the present study, we focused on the changes induced by distress on liver proteome. One of the liver functions is the regulation of the redox status. Although the hepatic antioxidant system is one of the most effective in the body, psychological distress can induce ineffective antioxidant defenses at this level. Our proteomic results showed several changes in proteins involved in the maintenance of redox homeostasis (Table S1). We observed changes induced by the distress in 19 proteins abundance from a total of 26 proteins described at this level, between them were the enzymes CAT, isoforms of GST, and SOD1. Also, enzyme activity was changed by oxidative stress in the liver, like the increase in GPX activity. This enzyme is involved in modulating cellular oxidant stress and redox-mediated responses (Lubos, Loscalzo & Handy, 2011). The GPX activity increase is probably due to the need of the cells to scavenge the lipid peroxides formed because of the increase ROS level due to psychological distress (Djordjevic et al., 2011), and piracetam performed its antioxidant function preventing this damage, as previously seen in immune (Grigoruţă et al., 2018) and neuronal cells (Gupta et al., 2014; Verma et al., 2018). In a CUMS rat model, an increase in oxidative proteins and lipids oxidation was observed in the liver after 40 and 60 days of stress (López-López et al., 2016) and, also, five days of psychological distress exposure altered proteins and lipids level in circulating immune cells (Grigoruţă et al., 2018). These alterations can be due to changes in RE and proteasome pathways involved in the modulation of cell response to high levels of glucocorticoids and catecholamines, as a response to stress (López-López et al., 2016). Moreover, several studies have found that distress promotes lipid peroxidation in the liver and other organs (Demirdaş, Nazıroğlu & , 2016; Duda et al., 2016; López-López et al., 2016). CUMS in male rats induced no changes in the SOD activity in the liver; however, CAT activity was significantly reduced after 20 days of stress (López-López et al., 2016). Besides, a chronic mild stress-induced hepatic oxidative stress model promoted the increase of ROS and lipid peroxidation and the decrease of GPX and CAT activity (Duda et al., 2016). The divergences among the different studies may be due to the type of the psychological stress model, the time of exposure to distress, and the gender-associated differences in response to oxidative stress (Brunelli et al., 2014).

Previous studies showed that piracetam decreases the antioxidant level, and therefore decreases the oxidative stress, and increases the proinflammatory responses (Singh et al., 2011; Liu et al., 2017; Verma et al., 2018). Moreover, this drug may protect the lipids from cellular membranes against oxidative stress due to its ability to scavenge free radicals, in this way avoiding mitochondrial dysfunction caused by such reactive molecules (Keil et al., 2006; Grigoruţă et al., 2018). In the present study, piracetam decreased the GPX and CAT activity in liver in distress conditions (Table 1) due to its ability to reduce the number of oxidant compounds, as observed in blood peripheral immune cells using SR-µFTIR (Grigoruţă et al., 2018).

However, in non-distressed conditions, possible alterations were shown in circulating mononuclear cells from rats treated with this drug (Grigoruţă et al., 2018). Piracetam has been labeled as a metabolic enhancer as it increases oxygen consumption and mitochondrial activity in stressed neuronal cells (Gupta et al., 2014; Verma et al., 2018). However, little evidence has been found about the effects of this drug on the metabolism under physiological conditions. In previous studies, piracetam has no impact on non-stressed subjects, both on the brain membrane fluidity of young animals (Müller et al., 1997) and cell viability and DNA damage in leukocytes and macrophages (Singh et al., 2011). In the present study, non-distressed groups showed changes in liver protein abundance and activity under treatment with this drug (Table 1 and Table S1). Our results suggest that piracetam induces an increase of catabolism rate guided to energy production, which can boost the ROS production and oxidative stress, under physiological conditions. These results support our previous observations made on circulating mononuclear cells from rats under the same distress conditions, in which a modest increase in oxidative stress in the control group (S−P+) was shown by using SR-µFTIR analysis (Grigoruţă et al., 2018). In the same way, the high fatty acid degradation observed in the present study can explain the decrease of lipids/proteins ratio reported in the immune cells from non-distressed rats, which confirms that this drug disturbs the metabolism (Grigoruţă et al., 2018). Therefore, the beneficial effect of piracetam seems to be closely related to the presence of oxidative stress (Keil et al., 2006; Verma et al., 2018).

Non-alcoholic fatty liver disease (NAFLD)

Recently, oxidative stress has been considered a factor that contributes to the NAFLD pathogenesis due to its effects on lipid metabolism. ROS overproduction in mitochondrial electron transport chain (ETC) and non-ETC ROS sources, as fatty acid β-oxidation, appears to contribute to hepatic metabolic diseases (Chen et al., 2020). Lipid peroxidation has been recently proposed as a possible mechanism for NAFLD development and progression because several circulating biomarkers of this process were identified in patients with NAFLD, as well as the presence of lipid peroxidation products (e.g., MDA and 4-HNE) were correlated with various histological features of non-alcoholic steatohepatitis (Chen et al., 2020). However, little evidence had been provided on the possible relationship between NAFLD and psychological distress. In a recent study, the effects of CUMS on metabolism were assessed in a mouse model of diet-induced obesity, and stress was not associated with changes in liver lipid deposition. However, a high-fat high-fructose diet promoted histological changes in the mouse liver that includes microvesicular lipid droplets, ballooning of hepatocytes, and cell infiltration (De Sousa Rodrigues et al., 2017). Our proteomic results may confirm that prolonged exposure to distress can lead to the development of NAFLD.

Parkinson’s disease (PD) (rno05012) was one of the enriched pathways. PD shares 86 gene entries with the NAFLD according to the KEGG pathway database (https://www.genome.jp/kegg/), which means that PD is associated with the oxidative stress and mitochondrial dysfunction induced by psychological distress (Grigoruţă et al., 2019; Grigoruţă et al., 2020), a molecular response that affects also other organs like the liver (López-López et al., 2016). Many proteins could be involved in the NAFLD progression. In this study, 38 proteins were found variable and belonging to the PD pathway, and 17 of these were not included in the NAFLD pathway (Table S2). One of these shared proteins is the PD protein 7 (PARK7) or protein DJ-1, which is a multifunctional protein with transcription modulatory and antioxidant activity, and acts as a cytoprotective in different cellular compartments (Junn et al., 2009). This protein is also required for the maintenance of mitochondrion homeostasis, like mitochondrial morphology and function, as well as for autophagy of dysfunctional mitochondria (Xiong et al., 2009). Loss of DJ-1 causes impaired mitochondrial respiration, increased intramitochondrial ROS, reduced mitochondrial membrane potential, and mitochondrial morphology alteration (Krebiehl et al., 2010). Moreover, DJ-1 is involved in the regulation of antioxidant gene expression through the Nrf2 transcription factor and of oxidative stress-induced apoptosis and has been related to the pathophysiology of immune and inflammatory diseases (Zhang et al., 2020). However, DJ-1 has an essential role in the progression of liver diseases due to its function in the inflammatory response initiation by modulating ROS generation and the immune response. In this study, DJ-1 abundance decreased by 50% in distress conditions. This observation shows the high use of this protein and the significant oxidative damage in the liver induced by psychological distress. A lower expression of DJ-1 was also found in the prefrontal cortex from rats exposed to the same model of distress (Grigoruţă et al., 2019) and in circulating immune cells from distressed PINK1-KO rats (Grigoruţă et al., 2020). In addition to the altered level of DJ-1, distressed rats showed anxiety, motor disfunction, altered mitochondrial respiration in the brain, and ineffective immune response (Grigoruţă et al., 2019; Grigoruţă et al., 2020). However, piracetam increased DJ-1 level in the liver, almost 80% in non-distressed animals, due to the metabolic enhancer effects of this drug. Nevertheless, a previous study showed that the lack of DJ-1 protected against hepatic steatosis because it enhanced the fatty acid oxidation that decreases the hepatic TG accumulation (Xu et al., 2018). Therefore, further studies are necessary for the role of this protein in the NAFLD progression in the liver, including gender-associated differences.

The connections of DJ-1 with members of the oxidative phosphorylation cluster have been reported in PD and other neurodegenerative diseases. This protein, along with others, assists the regulation of adequate ROS production in mitochondria by ETC and, therefore, maintains a proper mitochondrial function (Rekatsina et al., 2020). DJ-1 has been also related to the antioxidant enzymes CAT, GST, and SOD2 through the Nrf2 transcription factor, which regulates numerous antioxidant genes, being DJ-1 a positive regulator of Nrf2 (Yan et al., 2015).

ER stress

Distress and piracetam also induced alteration in protein processing in the ER. Both rough and smooth ER are widely developed in hepatocytes. The ER homeostasis alteration triggers the ER stress, which promotes the cumulation of misfolded or unfolded proteins in this organelle. ER stress has been associated with diverse pathophysiological changes, including inflammation, lipogenesis, insulin resistance, and apoptosis, both animal models and human subjects with NAFLD. The ER is also the origin of ROS, but their input into oxidative stress in NAFLD continues to be unclear (Chen et al., 2020). Our results suggest that distress exposure can induce an increase of the machinery to re-fold denatured proteins or to remove them from the ER-associated protein degradation pathway by the proteasome, probably to prevent the ER stress in the liver. In our previous study, the increase of protein denaturation or an abnormal protein aggregation was found in circulating mononuclear cells from stressed rats by SR-µFTIR analysis. Piracetam could not avoid such protein damage in the stress condition and it altered the proteins in the non-distressed group (S−P+), which was statistically grouped with S+P+ by principal component analysis (Grigoruţă et al., 2018). The ER stress has been associated with various liver diseases including NAFLD (Lebeaupin et al., 2018). Oxidative stress, lipotoxicity, and local inflammation induce the progression from hepatic steatosis to steatohepatitis and cirrhosis. To fight against cellular stress and inflammation, the UPR is a vital mechanism in patients with non-alcoholic steatohepatitis, especially in the mitochondria, because factors, such as mitochondrial UPR and mitohormesis, prevent the evolution of fatty liver to fibrosis (Solano-Urrusquieta et al., 2020). Previous studies showed that the PPAR signaling pathway (through PPARα receptor) can assist ER stress by regulating fatty acid β-oxidation pathways in peroxisomes and mitochondria, and also by inhibiting the inflammatory response in the liver (Wang et al., 2017). This pathway has been shown to mediate hepatic protection and the decrease of hepatocyte apoptosis, by regulating ER stress, in acute liver failure (Zhang et al., 2016). In this study, the PPAR signaling pathway was increased in stressed animals and by piracetam, which can confirm the ER stress.

Metabolism of xenobiotics by Cyt P450

The biotransformation process involving Phases I and II enzymes of xenobiotics and drug metabolism was also affected by distress in the present study. Similar results were found in a previous study, in which psychological stress alone impacts the regulation of liver proteins involved in Phase I and II, affecting the liver functionality, in a restraint stress mouse model (Flint et al., 2010). In our study, the cytochrome P450 2E1 (CYP2E1) was found more abundant in the rats’ liver after exposure to distress. This is one of the most abundant proteins in the liver of stressed rats and has been associated with steatohepatitis development. The functional stabilization of CYP2E1 induced hepatic oxidative stress, JNK1 signaling activation, insulin resistance, and accumulation of fatty acids and triglycerides, which results in liver injury, contributing to non-alcoholic steatohepatitis (Kim et al., 2016). This result supports the relationship between chronic exposure to psychological distress and the progress of NAFLD (Solano-Urrusquieta et al., 2020). In this study, piracetam avoided this change, but altered CYP2E1 expression in the liver of non-distressed rats, probably due to its stimulatory effect on metabolism (Müller et al., 1997; Keil et al., 2006; Kurz et al., 2010; Verma et al., 2018).

Effect of piracetam on animal behavior

Piracetam did not change the rat behavior under five days of exposure to distress, contrary to our previous study where nine days of drug administration diminished the anxiety behavior induced by distress (Grigoruţă et al., 2018), probably because a longer period of drug treatment is needed to obtain significant results. However, the drug induced a change in animal conduct when it was administered under physiological conditions. This result confirms our previous studies, where piracetam induced neurostimulation and psychomotor agitation in healthy individuals (Corazza et al., 2014; Grigoruţă et al., 2018), probably due to its actions as a metabolic enhancer (Müller et al., 1997; Keil et al., 2006; Kurz et al., 2010; Verma et al., 2018).

Effect of distress and piracetam on biochemical parameters

In the present study, none of the specific markers for liver injury, like ALT and AST, were affected by distress and/or piracetam, although the total bilirubin tended to decrease, regardless of whether rats were exposed or not to distress. Low bilirubin levels have been closely related to cardiovascular diseases or a high risk of cerebral deep white matter lesions (Higuchi et al., 2018). Contrary to our results, CUMS model induced an increase in the activity of ALT and AST in male rats, suggesting liver damage (Jia et al., 2016). Again, gender-associated studies need further investigations.

Related to specific markers associated with renal function, the exposure to distress induced a significant increase in BUN level, indicating that the sympathetic system activation by psychological distress stimulates protein degradation, or/and induces hemodynamic changes, that can lead to kidney function failure and increase the risk for cardiovascular disease (Kazory, 2010). Piracetam protected kidney functions by diminishing BUN and urea levels. Previous studies showed the beneficial effect of this drug on microcirculation not only in the brain but also in the periphery (Winblad, 2005).

Conclusions

In the present study, we verify that our emotional distress paradigm causes changes in proteins involved in redox status homeostasis, ER stress, and the biotransformation processes in the distressed rat liver. These results support the relationship between the distress and the development of NAFLD and non-alcoholic steatohepatitis. We propose new potential targets of interest in liver-related as the antioxidant proteins DJ-1, GPX, PRDX5, GLRX5, TXNDR1, and plasma biochemical parameters as urea and BUN levels related to kidney function for further studies.

Piracetam appears to counteract the molecular effects caused by distress in the liver, although it has the opposite effect under normal conditions. Thus, we confirmed that piracetam has antioxidant effects on the liver as long as the body is subjected to the molecular effects of psychological distress.

On the other hand, the influence of gender in the physiological response to oxidative stress and how this can affect the body and evolve in diverse pathologies should be further studied since females seem to better counteract the effects of distress on the brain, as previously described (Grigoruţă et al., 2019).

Finally, these results generate new hypotheses for the molecular mechanism study of how psychological distress affects liver function and new pharmacological applications of known drugs in the whole organism protection.

Supplemental Information

Supplemental Information 1 Additional information on the methodology used in the GeLC-MS/MS analysis

A detailed description of the in-gel digestion, the nanoLC-MS/MS analysis, and the protein search parameters are included.

Click here for additional data file.

Supplemental Information 2 Experimental design

For six consecutive days, rats received piracetam (600 mg/kg), dispensed orally using a cannula, in single dose, at 7:30 am. From the first day until the fifth day, the animals were exposed to the stressor (at 8 am, after drug administration). In the sixth day, rats were sacrificed.

Click here for additional data file.

Supplemental Information 3 Correlation between behavior, plasma biochemical parameters, and GPX activity in rats exposed to psychological distress and piracetam

The glucose level was negatively correlated with the time the rat spent hiding (A) and positively with the head-out position (B). The direct bilirubin level was positively correlated with the time the rat spent approaching (C) and exploring (D). The total bilirubin level was negatively correlated with the time the rat spent hiding (E) and positively with the exploring time (F). The GPX activity was positively correlated to the time the rat spent hiding (G) and negatively with the exploring time (H).

Click here for additional data file.

Supplemental Information 4 Fold change in the abundance of redox proteins identified by GeLC-MS/MS analysis in the different conditions

S−P−: unique protein detected in S−P − condition; S+P−: unique protein detected in S+P− condition; S+P+: unique protein detected in S+P+ condition.

Click here for additional data file.

Supplemental Information 5 Identified proteins involved in non-alcoholic fatty liver disease (NAFLD) and Parkinson’s disease (PD) pathways according to KEGG database

The fold change was obtained from the ratio between the relative protein abundance based on the emPAI value of the indicated groups. S−P−, S+P−, S+P+, S −P+ indicate proteins that were detected only in S −P−, S+P−, S+P+, or S−P+ group respectively. n.d. indicates protein that was not detected in the experimental group. PD and both indicate proteins that are categorized only in PD pathway or in both pathways, respectively. S−P −, rats neither exposed to stress nor with piracetam treatment; S−P+, rats not exposed to stress but treated with piracetam; S+P−, rats exposed to stress without piracetam treatment; S+P+, rats exposed to stress and treated with piracetam.

Click here for additional data file.

Supplemental Information 6 Proteins (indicated by their protein symbol) that showed changes of 50% in protein abundance and are involved in protein processing in ER, proteosome, and metabolism of xenobiotics by Cyt P450

Most abundant indicates proteins with fold change (FD) ≥1.5. Less abundant indicates proteins with FD ≤0.66. n.d. indicates protein that was not detected in the experimental group.

Click here for additional data file.

Supplemental Information 7 Pearson’.s correlation among treatment, biochemical parameters, antioxidant enzyme activities, and rat behaviors

Significant values are marked in bold (* p ≤ 0.05; †p ≤ 0.01).

Click here for additional data file.

Supplemental Information 8 Raw data collected in the GeLC-MS/MS analysis from rat liver proteins for all the treatments (S−P−, S+P−, S+P+, and S−P +)

The data contain details of identified proteins: Accession number, Description, Coverage, # Peptides, # Unique Peptides, MW [kDa], calc. pI, emPAI, Score Sequest, Coverage Sequest, # Peptides Sequest, Score Mascot, Coverage Mascot, # Peptides Mascot, Score MS Amanda, Coverage MS Amanda, # Peptides MS Amanda, and identification data for each detected peptide.

Click here for additional data file.

Supplemental Information 9 Fold change in the abundance of liver proteins identified in the different experimental conditions

Proteins with changes in abundance of 50% were selected ( 0.66 ≤ fold change ≤1.5). The UniProtKb number accession, protein description, and fold change are showed. S−P−, S+P−, S+P+ or S−P+ in the ratio column indicates proteins only detected in a particular condition. S−P −, rats neither exposed to stress nor with piracetam treatment; S−P+, rats not exposed to stress but treated with piracetam; S+P−, rats exposed to stress without piracetam treatment; S+P+, rats exposed to stress and treated with piracetam.

Click here for additional data file.

Supplemental Information 10 Data collected from the KEGG pathway enrichment analysis using Metascape resource

The settings used were p-value < 0.01, minimum count of 3, enrichment factor > 1.5 and FDR > 0.05. Data show the GO pathway identifier, pathway name, log10 (p-value), log10 (q-value), gen number of upload hit list in each pathway/gene number of genome in the same pathway, list of Entrez Gene IDs of upload hits in the pathway, and the list of symbols of upload hits in the pathway.

Click here for additional data file.

Supplemental Information 11 Raw data from the antioxidant enzyme activity, plasma biochemical parameters, and rat behavior analyses

The individual data that were obtained in the different analyses are shown for each experimental group.

Click here for additional data file.

Supplemental Information 12 Data collected from the enrichment clustering analysis using the STRING protein-protein interaction database

Data collected from the enrichment clustering analysis using the STRING protein-protein interaction database. Proteins were clustered using the Markov Cluster Algorithm (MCL) with an inflation parameter of 3. Data show the cluster number and color, gene count, and protein name, identifier, and description.

Click here for additional data file.

Supplemental Information 13 ARRIVE guidelines 2.0 checklist

Click here for additional data file.

We are very grateful to Dr. Diana Marcela Beristain-Ruiz from Veterinary Science Department, Biomedicine Science Institute at the UACJ, for her invaluable support in the analysis of the biochemical parameters.

Additional Information and Declarations

Competing Interests

Author Contributions

Animal Ethics

Data Availability

The authors declare there are no competing interests.

Raquel González-Fernández conceived and designed the experiments, analyzed the data, prepared figures and/or tables, authored or reviewed drafts of the paper, and approved the final draft.

Mariana Grigoruţă conceived and designed the experiments, performed the experiments, authored or reviewed drafts of the paper, and approved the final draft.

Sarahi Chávez-Martínez performed the experiments, prepared figures and/or tables, and approved the final draft.

Eliel Ruiz-May and José Valero-Galván analyzed the data, authored or reviewed drafts of the paper, and approved the final draft.

José Miguel Elizalde-Contreras performed the experiments, authored or reviewed drafts of the paper, and approved the final draft.

Alejandro Martínez-Martínez conceived and designed the experiments, authored or reviewed drafts of the paper, and approved the final draft.

The following information was supplied relating to ethical approvals (i.e., approving body and any reference numbers):

The Institutional Committee of Ethics and Bioethics from the Universidad Autónoma de Ciudad Juárez, Ciudad Juárez, Chihuahua, Mexico provided full approval for this research (No.CIBE-2017-1-34).

The following information was supplied regarding data availability:

The raw measurements of the proteomic analysis performed by GeLC-MS/MS and raw data obtained in the antioxidant enzyme activity, plasma biochemical parameters, and rat behavior analyses are available in the Supplementary Files.

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
