# Peer review of "Liver proteome alterations in psychologically distressed rats and a nootropic drug"

_PeerJ, doi:10.7717/peerj.11483_

## Round 0.1 · original submission · Major Revisions

Thank you for submitting this interesting study to PeerJ. As suggested by both reviewers this study has its novelty; however, there are substantial issues that need to be addressed. It was found large variations in the control group and was suspected it might be due to female rats used. Please address all of the issues proposed and revise the manuscript with point-to-point responses.

Reviewer 1 ·

Basic reporting

Proteomics was deployed to study the effect of psychological distress and piracetam on the liver proteome. The manuscript was well structured and provided sufficient background and context.

Experimental design

I have a major concern about the experiment design for proteomics: liver samples were pooled for each experimental group, so there is no information about the variation in the group and between groups.

Validity of the findings

Data in Table 1 need to be verified.

Additional comments

Proteomics was deployed to study the effect of psychological distress and piracetam on the liver proteome. The manuscript was well structured and provided sufficient background and context, however, I have a major concern about the experiment design for proteomics: liver samples were pooled for each experimental group, so there is no information about the variation in the group and between groups. Data in Table 1 need to be verified.

·

Basic reporting

- The professional English used in the article is clear.

- The introduction and background context of the manuscript is good, but I have the following observations:

- They emphasize the effect of psychosocial stress and relate it to neuroimmunological changes, highlighting various brain structures, but they will evaluate the effect of the drug on the liver, an organ that should be given more emphasis, since they will not be working with the brain. The authors comment that there is not enough information on this subject, but in the discussion they present a couple of articles that show liver damage after subjecting mice to stress. In addition, in the conclusions in the lines 705-6, they describe that the proteomic analysis of the liver from stressed rats supports the fact that emotional stress causes an imbalance in the metabolism of lipids and antioxidant enzymes.

- What is the difference between stress and distress? These two terms are used interchangeably in various sections of the article, mainly in the introduction, methodology, discussion and conclusions. It is convenient to handle the terms correctly since they are the parameters of your study. I suggest to review the article by Kiersten et al., 2021. Health Psychol. 2021 Feb;26(2):312-318. doi: 10.1177/1359105318804865.

- Authors of the manuscript indicate in the line 97 that Piracetam was the first of these drugs to be described and is commonly used as a neuronal enhancer because it mainly modulates cholinergic and glutamatergic, But Verma et al., 2018, showed that Piracetam is a cyclic derivative of the neurotransmitter γ-aminobutyric acid (GABA) and it is used in clinical practices for the treatment of epilepsy. Therefore, it is not clear if the drug decreases or increases neuronal activity, it is necessary to describe the mechanism of action more precisely.

- The cited literature is adequate and up-to-date. Although the following points must be corrected:

- The bibliographic citations present errors in the title of the journal, which in some cases appears as abbreviations and in others the complete name, it is necessary to homogenize them. For example the first reference: Abdel-Salam et al., 2011. The Journal appear as Neurochem Res, but in the third reference of Adachi et al, 1993, it says Cancer Research.
- A reference has an asterisk in the line 972.

- Missing reference of Jia et al., 2018. Include it.
-
- Place genus and species in italics and other words in latin language, such as in vitro and et al.

- Several references contain the title of the article in upper and lower case.

- The structure of the article conforms to PeerJ standard

- Figures:
o For figures 1 and 4 it is recommended to use contrasting colors in the graphs, do not highlight the colors used.

o The figure legends should highlight the results shown and not be limited to describe the experimental conditions.

Experimental design

- The research is original and with good impact. The research question is not completely clear.

- Molecular biology techniques are well described, and the research group has shown their management through several publications.

- I recommend a flow chart showing the experimental conditions used and a picture of the device they used for distress induction.

- They used female rats and did not find significant differences in their model, due to the large dispersion of the results in the control group compared to the stress group, which can be attributed largely to the number of animals and variations in their estrous cycle. I suggest documenting this part, since they comment that they have previous studies in males where they did observe differences. It is essential to validate their study model for the results obtained in the quantified proteins.

- Did you evaluate the estrous cycle of the rats? Since several research groups have identified changes in activity and rest throughout their cycle. In which part of the estrous cycle and the light-dark cycle were the tests performed?. It has been shown that various neurobiological parameters such as stress, sleep, learning, memory and sexual behavior vary according to the phase of the estrous cycle. I recommend reviewing the article by Oyola et al., 2017 (Oyola MG, Handa RJ. Hypothalamic-pituitary-adrenal and hypothalamic-pituitary-gonadal axes: sex differences in regulation of stress responsivity. Stress. 2017 Sep;20(5):476-494. doi: 10.1080/10253890.2017.1369523) to enrich the discussion of the obtained behavioral data.
- The statistical analysis performed is adequate.

Validity of the findings

- The manuscript presents an adequate discussion by sections, but I observe the following drawbacks:

- In the Introduction they point out the importance of the study performed but in a general way. I suggest highlighting the relevance of the study.

- In the section of Redox metabolism they point out that: The relationship between psychological stress and the redox status alteration has been previously shown in several studies using different murine models in the liver, brain, pancreas, kidney,lungs, and heart (Şahin & Gümüşlü, 2007; Jafari et al., 2014; Mejia-Carmona et al., 2014, 2015;López- López et al., 2016; de Sousa Rodrigues et al., 2017). After this paragraph, there is no description of the contribution of your study, it seems that it has already been done.

- They do not highlight the justification of the work. Therefore, their discussion can be enriched by considering the following aspects:

- In the previous studies performed by the working group they comment that protein expression changed, but it is convenient to specify which alterations or changes were presented.

- They also describe that Piracetam decreases the antioxidant level and therefore decreases the oxidative stress and increases the proinflammatory responses (Singh et al., 2011; Liu et al., 2017; Verma et al., 2018). Moreover, this drug may protect the lipids from cellular membranes against oxidative stress due to its ability to scavenge free radicals, in this way avoiding mitochondrial dysfunction caused by such reactive molecules (Keil et al., 2006). In our study, piracetam decreased the GPX and CAT activity in distress conditions due to its ability to reduce the number of oxidant compounds. So, they do not provide new information and only corroborate previously described results?

- In the Line 693 described: Contrary to our results, CUMS model induced an increase in the activity of ALT and AST in male rats, suggesting liver damage (Jia et al., 2016). Again, gender-associated studies need further investigations. It is not clear, What were your results?

- The conclusions are general, so they are incomplete to a large extent, it is necessary to highlight the results obtained. They comment that Piracetam had protective effects in the stress group, but not in relation to the controls and that this may be due to the fact that they used female rats, which in comparison with males are more resistant, the great inconvenience is that significant changes were registered only in the stress plus piracetem group in the behavioral test.

Additional comments

- One of the great advantages of the article is the large number of techniques used to evaluate stress-induced liver damage and the protective effect of Piracetam in rats. These techniques have been developed by the research group in other publications, and they only require to specify some aspects that I pointed out in each of the sections requested in the revision of the manuscript. Besides emphasizing the importance of the study and highlighting the results obtained.

---

## Round 0.2 · accepted · Accept

Thank you for submitting this interesting study to PeerJ. After a comprehensive review by the reviewers and editor, we thought this manuscript can be accepted for publication and this proteomic study in the liver could help us better understand molecular cellular processes under psychological stress.

·

Basic reporting

No comment

Experimental design

No comment

Validity of the findings

No comment